# Living Landmarks: A Review of Monumental Trees and Their Role in Ecosystems

**DOI:** 10.3390/plants14132075

**Published:** 2025-07-07

**Authors:** Ruben Budău, Claudia Simona Cleopatra Timofte, Ligia Valentina Mirisan, Mariana Bei, Lucian Dinca, Gabriel Murariu, Karoly Alexandru Racz

**Affiliations:** 1Department of Silviculture and Forestry Engineering, University of Oradea, 26 General Magheru Boulevard, 410048 Oradea, Romania; 2Department of Law and Administrative Sciences, University of Oradea, 26 General Magheru Boulevard, 410048 Oradea, Romania; claudia.timofte@uoradea.ro (C.S.C.T.); ligiamirisan@yahoo.com (L.V.M.); 3Faculty of Environmental Protection, University of Oradea, 26 Gen. Magheru Street, 410048 Oradea, Romania; mbei@uoradea.ro; 4National Institute for Research and Development in Forestry “Marin Dracea”, Eroilor 128, 077190 Voluntari, Romania; 5Department of Chemistry, Physics and Environment, Faculty of Sciences and Environmental, Dunarea de Jos University Galati, Domneasca Street No. 47, 800008 Galati, Romania; gmurariu@ugal.ro; 6PhD Student of the Doctoral School of Engineering Sciences, Field of Agriculture, Faculty of Environmental Protection, University of Oradea, 24 Gen. Magheru Street, 410048 Oradea, Romania; alex.racz@yahoo.com

**Keywords:** bibliometric analysis, *Olea europaea*, ecosystem services, monumental trees, champion trees

## Abstract

Monumental trees, defined by their exceptional size, form, and age, are critical components of both cultural heritage and ecological systems. However, their conservation faces increasing threats from habitat fragmentation, climate change, and inadequate public policies. This review synthesized global research on monumental trees by analyzing 204 peer-reviewed articles published between 1989 and 2024 that were sourced from Web of Science and Scopus. Our bibliometric analysis highlighted *Olea europaea* and *Castanea sativa* as the most frequently studied species and identified a surge in publications after 2019, particularly from the USA, Italy, and Spain. Key research themes included conservation, biodiversity, and ecosystem services. The methodological approaches varied globally, encompassing ranking systems; GIS mapping; remote sensing; and non-invasive diagnostic tools, such as acoustic tomography and chlorophyll fluorescence. Conservation strategies discussed included vegetative propagation, cryopreservation, and legal risk management. Despite advances in these techniques, significant gaps remain in effectively addressing environmental pressures and integrating multidisciplinary approaches. We concluded that targeted, interdisciplinary strategies are essential to safeguard monumental trees as vital ecological and cultural landmarks.

## 1. Introduction

Trees are integral components of ecosystems, whether they form parts of forest stands or exist as isolated specimens outside forests. Their ecological role includes influencing and regulating various environmental factors, such as the microclimate, air quality, soil stability, and biodiversity [1,2,3,4,5].

The inclusion of remarkable historic trees in urban ecosystems represents significant ecological assets, with potential contributions to the regulation of environmental conditions [6]. However, on a broader scale, this natural and cultural heritage remains insufficiently understood, and its adequate conservation is increasingly threatened in the urban areas of developed countries [7].

A variety of terms are used in the scientific literature to refer to old and significant trees. These include ancient, veteran, notable, champion, old, monumental, heritage, and mother trees [8,9]. Each of these terms highlights specific attributes: exceptional size (champion), cultural value (heritage) [10], ecological importance (large old trees) [11], or structural complexity and age (ancient, veteran) [12,13].

Heritage trees embody and symbolize culture, tradition, and human history, offering substantial aesthetic, religious, and landscape values [14].

Monumental trees can generally be categorized into four main groups: historical, mystical, folkloric, and dimensional. As the name suggests, dimensional monumental trees are defined by their distinctive size, species, and habitat, typically being over 100 years old [15]. Nevertheless, these categories should not exclude other trees with notable ecological or cultural value, which may still warrant protection [16]. Monumental trees can be distinguished from common specimens by features such as exceptional age, height, trunk diameter, crown structure, form, and the presence of associated legends or historical significance [17].

An analysis of the composition, performance, and growth outlook of historic trees in the urban forest of Guangzhou, China [7], assessed the floristic composition, age profile, and biomass structure to evaluate the tree performance across key habitats (institutional grounds, parks, and roadside areas). The study identified limited common characteristics, indicating that the tree performance varies with habitat, thereby exerting selective pressure on species.

In Guangzhou’s urban forest, few trees exceeded 300 years in age, reflecting a historical pattern of conflict between urban development and old trees. While the trunk and crown dimensions suggest considerable potential for future biomass expansion and visual impact, such potential appears constrained by the continuous growth of the urban infrastructure [7,15].

Across cultures, humans have long revered large and old trees, often associating them with mythology, folklore, historical events, and artistic representation. These associations underline both the economic and cultural significance of monumental trees [18].

Urban green spaces provide cultural, aesthetic, and ecological services and may play a crucial role in preserving the native plant diversity in cities [19]. An investigation into the coexistence of people and nature—based on the composition of heritage tree species and perceived cultural values—in the Yangtze River basin, encompassing 11 major cities, revealed close links between heritage trees and human–nature relationships. However, the species composition varied by region, and comparisons between heritage trees, ornamental urban trees, and natural vegetation showed distinct differences [20,21].

Heritage trees are important for historical, scientific, and commemorative purposes [22]. Generally, trees over 100 years old are rare or endangered species, are located at culturally significant sites, or were planted by notable figures. As such, these trees contribute to the richness of rural and national landscapes. Memorial trees, in particular, are designated by conservation authorities and protected by law due to their age, physical characteristics, location, or symbolic significance. These attributes make them key elements of both cultural and environmental narratives [23].

The aesthetics of trees and urban vegetation significantly shape urban landscapes, directly impacting residents’ quality of life and the perception of public spaces [24]. Activities related to the maintenance and development of green heritage, typically overseen by municipal green space departments, are crucial for promoting sustainability and urban resilience [25].

Research on the divergent phenological responses of plant species to seasonal temperature in the temperate forests of Mount Doling, Beijing, demonstrated that asymmetric seasonal warming can significantly impact plants and ecosystems as a whole [26].

In this article, we use the term monumental trees as the most appropriate and encompassing descriptor for these various aspects.

Our review revealed that although numerous studies have been published on various plant species [27,28,29] and forest ecosystems [30,31,32,33], there is no comprehensive article focused exclusively on monumental or champion trees. Therefore, the primary goal of this study was to review the existing scientific literature regarding monumental trees, using both classical and bibliometric approaches.

The objectives of this study were to (1) conduct a bibliometric analysis of the scientific literature on monumental and champion trees between 1989 and 2024; (2) identify publication trends, geographic coverage, research domains, authorship, and keywords; (3) highlight the most frequently cited monumental tree species, along with their locations and cultural significance; (4) review the global methods used to inventory and classify monumental trees, including size ranking, remote sensing, and GISs; (5) analyze current conservation strategies and legal frameworks for protecting monumental trees in both rural and urban environments, with emphases on risk assessment, propagation, and public engagement; and (6) identify knowledge gaps and propose future directions for integrating the conservation of monumental trees within ecological, cultural, and technological frameworks.

## 2. Results and Discussion

### 2.1. A Bibliometric Review

Regarding this topic, up until 2025, 204 publications were identified. The distribution of these was as follows: 151 articles (75% of total publications), 28 proceeding papers (14%), 16 review articles (8%), and 7 book chapters (3%) (Figure 1).

The published articles could be classified into 77 scientific fields. Among these, the ones with the highest number of articles were *Environmental Sciences and Ecology* (59 articles), *Forestry* (46 articles), and *Agriculture* (41 articles) (Figure 2).

The annual distribution of the published articles (Figure 3) shows that their number increased significantly after 2019.

Many authors published between one and five articles on this topic. Leading the list was Margot Dudkiewicz with five articles, followed by Pawel Zarzynski and Marek Dabski, each with four articles.

Researchers from 64 countries on five continents contributed to articles on this topic (Figure 4). The most represented countries were the USA (82 articles), Italy (56 articles), China (41 articles), Spain (30 articles), and Poland (27 articles).

It was observed that countries were grouped into several clusters, three of which each included five countries: cluster 1: Austria, Cyprus, Germany, Greece, and Spain; cluster 2: Australia, Czech Republic, England, Scotland, and Switzerland; cluster 3: Nigeria, Pakistan, China, South Africa, and South Korea (Figure 5).

The majority of articles are written in English, but there were also articles in nine other languages, namely, Polish, Spanish, Turkish, French, Russian, Portuguese, German, Greek, and Italian.

Articles on this topic were published in 289 journals, with the most articles published in *Sylvan* (eight articles), *Sustainability* (seven articles), and *Forests* (six articles) (Figure 6).

The most representative publishers were MDPI (23 articles), Elsevier (20 articles), Springer (14 articles), and Wiley (8 articles).

The most frequently used keywords were *conservation*, *Olea europaea*, *monumental trees*, and *management*, based on the number of articles. Based on the total link strength, the most frequent keywords were *conservation*, *biodiversity*, and *Olea europaea* (Table 1).

The keywords could be grouped into several clusters, of which two contained at least 10 keywords. These were as follows: the first cluster, which included the keywords dendrochronology, dynamics, ecosystem services, evolution, forestry, growth, monumental trees, performance, phylogeny, trees, urban forest, and vegetation; the second cluster included the keywords age, ancient trees, cultivated olives, domestication, ecology, efficiency, genetic diversity, identification, olea europaea, quality, and wild (Figure 7).

Between 2015 and 2018, the predominant keywords used in the published articles were monumental tree, urban forest, growth, and evolution; during 2019–2020, these were tree, biodiversity, conservation, and management; while during 2021–2023, the predominant keywords were landscape, benefits, history, and *Olea europaea* (Figure 8).

The temporal evolution of keywords provides both theoretical and practical insights into how the field of monumental tree research has developed over time. Theoretical significance lies in tracing the emergence and decline of core concepts, revealing shifts in the academic focus—such as a transition from growth and taxonomy to conservation and ecosystem services. Practically, this analysis helped identify trends that align with broader environmental policy developments (e.g., the rise of “biodiversity” and “management” after 2019) and highlighted underexplored or emerging topics that may warrant further investigation. By recognizing these shifts, researchers and policymakers can better align future studies and funding priorities with current environmental and societal needs.

### 2.2. Literature Review

#### 2.2.1. Examples of Monumental Trees

Following a thorough analysis of the articles published on this topic, we identified the following species of trees mentioned as monumental or champion trees (Table 2).

Monumental or champion trees are widely distributed across various continents, climates, and ecosystems, reflecting both their ecological adaptability and cultural significance. Table 2 presents a curated selection of 45 tree species identified in the scientific literature as being notable for their exceptional age, size, or historical importance.

In our bibliographic review, we identified 45 tree species that include individuals recognized as monumental trees in various countries worldwide. However, this number likely underrepresents the true diversity, as nearly any of the approximately 73,300 known tree species globally may have specimens that qualify as monumental trees.

These species range from temperate zone conifers, such as *Abies alba* and *Picea abies*, to tropical and subtropical representatives, like *Ficus microcarpa* and *Taxodium mucronatum*. Several species, including *Castanea sativa*, *Olea europaea*, and *Quercus robur*, appear in multiple geographical regions, indicating both their broad native ranges and long-standing human association, often due to their roles in agroforestry or cultural practices.

Among the tree species frequently cited in the scholarly literature, olive trees (*Olea europaea*) stand out as the most commonly mentioned. In the Mediterranean region, they have long been revered for their exceptional longevity. Their substantial size has often fueled claims of great antiquity, with many specimens referred to as “ancient olive trees” and believed—sometimes controversially—to be as old as 2000 years. A major cultivated species throughout the Mediterranean, *Olea europaea* is often represented by large, gnarled trees thought to date back millennia [80,81]. Recent studies have identified ancient specimens, both wild and cultivated, across several Mediterranean countries, including Italy [82,83], Spain, Greece [84], Montenegro [85], Israel and the Palestinian territories [86,87], and Tunisia [88]. These venerable trees serve as living testaments to the region’s long history of olive cultivation and exemplify the species’ remarkable resilience and longevity under challenging environmental conditions [89].

The chestnut tree (*Castanea sativa*) is the second-most frequently referenced species in the literature concerning monumental trees. Cultivated since ancient times for its multifaceted utility—ranging from timber and edible nuts to honey production and tannins—it has more recently been appreciated for its ecological contributions, particularly in stabilizing forest ecosystems and mitigating natural hazards [90]. *Castanea sativa* has long been intertwined with human activity and rural economies across Europe, especially since the Middle Ages. Widely grown both in orchards for food and in coppices for timber, it became a cornerstone resource for many mountain communities [91]. In addition to its economic and environmental significance, the species’ leaves and flowers have held an enduring place in traditional medicine [92].

Certain regions stand out for their densities of monumental trees reported in the literature. For instance, Romania, Poland, and Turkey frequently recur in the table, highlighting both the presence of well-preserved old-growth stands and the national-level interest in dendrological heritage. Urban monumental trees are also represented, such as *Ailanthus altissima* and *Catalpa bignonioides*, underscoring the role of cityscapes in preserving notable specimens.

A notable aspect is the presence of both native and introduced species, with examples such as *Ginkgo biloba* and *Sequoiadendron giganteum* being celebrated far outside their original habitats. This reflects a broader trend of transcontinental arboreal admiration and the symbolic value attached to these species, often leading to deliberate conservation efforts.

Moreover, some tree genera (e.g., *Tilia*, *Quercus*, *Taxus*) are represented by more than one species or subspecies, indicating that monumental status can be achieved across a genus, depending on the environmental conditions and historical land use.

Overall, this diversity of monumental trees underscores their ecological, historical, and cultural importance, providing not only structural complexity in their ecosystems but also serving as living records of past human–nature interactions.

#### 2.2.2. Methods for Inventorying Monumental Trees

Across the globe, monumental trees are identified and recorded through a range of national and regional inventory systems that emphasize not only the tree size, but also the cultural significance and ecological value. For example, in the United States, the National Register of Champion Trees is maintained by *American Forests*, a nonprofit conservation organization [93]. Champion trees are ranked using a point system based on the trunk circumference (in inches), total height (in feet), and average crown spread (in feet), and must be recertified every ten years. Many U.S. states have their own independent lists, usually managed by state departments of natural resources and based on similar criteria [94,95]. The North Carolina Forest Service’s Big Tree Program, initiated in the late 1960s, is one such registry. While it historically relied on public nominations and local arborist input, the database has suffered from inconsistent data reporting and infrequent updates and is often only modified when a larger tree is discovered or an existing one is lost—limiting its current utility [96].

Comparable systems exist elsewhere. In Canada and the United Kingdom, champion trees are similarly ranked by size. South Africa’s Champion Tree Register includes measurements such as the diameter, height, and crown spread, while also incorporating the tree age and heritage value. In Hong Kong, evaluation factors include the age, form, health status, dimensions, rarity, ecological role, aesthetic value, and historical associations [97]. In Ireland, the *Tree Register of Ireland (TROI)*—founded in 1999 by the Irish Tree Society and the Tree Council of Ireland—catalogs “champion trees” based on their size, age, rarity, or cultural relevance, and has documented over 14,000 specimens to date [98].

In Chile, monumental trees are recognized for their ecological and cultural contributions—such as promoting the public appreciation of nature, supporting biodiversity, and serving as biocultural heritage. Gutierrez proposed the development of a national inventory and policies to safeguard these trees [99].

Germany has its own registry through the German Dendrological Society (DDG), with prominent publications like those by Kűhn et al. [100]. Since 2009, a joint initiative with the *Society German Arboretum (GDA)* has maintained an online, continuously updated register. Since 2010, the registry has featured an annual “Champion Tree of the Year,” which is promoted through public ceremonies [101].

At the international level, Hong Kong, New Zealand, and other countries contribute data to broader databases. The *European Tree of the Year* contest, which stems from national-level competitions, raises the public awareness of iconic trees. The *European Champion Tree Forum (ECTF)*, founded in 2010, promotes collaboration across borders. The globally accessible *Monumental Trees* online database offers public data on over 30,000 notable trees [102].

In Turkey, until recently, there was no standardized approach for identifying monumental trees. A new method now allows for structured selection based on tree characteristics, including the estimated age, total height, diameter at breast height (DBH), crown diameter, and site quality. Data are gathered through oral questionnaires, allowing participants to score positive and negative traits that influence a tree’s monumental value [103].

Beyond traditional measurements, technological innovations have significantly improved the inventory accuracy and tree health assessments. In Turkey, field-based measurements (crown diameter, trunk diameter, and height) have been integrated with Geographic Information Systems (GISs) to develop spatial tree databases, enabling statistical and geostatistical analyses for improved conservation planning [104].

In Italy, the “From Space to Tree” (S2T) method combines multi-scale, multi-sensor remote sensing to monitor vegetation in historical and archaeological parks. This technique assesses the vegetation health, biomechanical stability, and risks at the vegetation–monument interface [105].

In Poland, a set of non-invasive diagnostic tools—including acoustic tomography, electrical tomography, and chlorophyll fluorescence analysis—has been used to evaluate tree health [106]. Devices like the Resistograph, which assesses internal decay, are commonly used in similar contexts [34,107].

A more novel approach in the Pacific Northwest involves species distribution modeling with Maxent, which integrates ecological field data and archaeological information to predict the locations of monumental western redcedar (*Thuja plicata*)—a culturally important species for Indigenous communities that is traditionally used for canoes and totem poles, which is now in decline due to logging [78].

In Italy’s Madonie Park, researchers applied the Contingent Valuation Method (CVM) to quantify the “existence value” of monumental trees along a nature trail that includes giant hollies and other notable specimens [108]. In the Apulia region, monumental ancient olive groves were assessed for cultural significance through trunk dimension measurements and visual trunk shape analysis, as mandated by Regional Law 14/2007 [109].

These global practices demonstrate the increasing complexity and interdisciplinarity of monumental tree inventories. While many systems continue to rely heavily on size-based rankings, others—like those in Chile, South Africa, and Hong Kong—incorporate broader ecological, cultural, and historical criteria. The trend toward public engagement (e.g., the European Tree of the Year contest), open-access platforms (like Monumental Trees), and international collaboration (e.g., ECTF) reflects a shift toward more inclusive and dynamic conservation frameworks.

At the same time, technological tools are reshaping how inventories are conducted. GIS, remote sensing, and tomographic analyses allow for rapid, non-destructive assessments of health, structure, and risk. Tools like the Resistograph and chlorophyll fluorescence analysis are particularly crucial in managing aging trees. Meanwhile, economic valuation methods, such as CVM, and culturally responsive frameworks, like those involving visual assessments of ancient olive trees, help quantify non-material values associated with monumental trees.

Nonetheless, these trees face mounting threats from climate change, drought, disease, and urban expansion. Static registries alone are insufficient. To ensure their preservation, inventory systems must evolve into adaptive, living frameworks—serving not just science, but also education, culture, and policy. A comprehensive, interdisciplinary approach is essential for sustaining these “living landmarks” for future generations [8,10,110,111,112,113,114].

#### 2.2.3. Guidelines and Methods for the Conservation of Monumental Trees

##### Inventory Approaches: National and Regional Models

Across various countries, monumental trees are identified and recorded through national registries and conservation programs, typically emphasizing tree size, cultural significance, and ecological value. In the United States, the National Register of Champion Trees is maintained by American Forests, a nonprofit conservation organization [80]. Champion trees are ranked using a point system based on trunk circumference (in inches), total height (in feet), and average crown spread (in feet). These measurements must be updated and recertified every ten years. Many U.S. states also maintain independent champion tree lists, typically managed by state departments of natural resources, using similar criteria [81,82].

For example, the North Carolina Forest Service (NCFS) maintains the Big Tree Program, an inventory system for champion-sized trees based on trunk circumference, height, and crown spread. Initiated in the late 1960s, this program has historically relied on public nominations, often in collaboration with local foresters and arborists. However, due to inconsistent data reporting and infrequent updates—typically only when a larger specimen is found or an existing one is lost—the accuracy and utility of the register have diminished over time [83].

Comparable systems exist in Canada and the United Kingdom, where champion trees are ranked using similar size-based metrics. In South Africa, the Champion Tree Register incorporates not only tree dimensions (diameter, height, and crown spread) but also age and heritage significance. In Hong Kong, evaluation criteria include the tree’s age; form; health status; dimensions; and attributes such as species rarity, ecological role, aesthetic value, and historical associations [84].

Internationally, several models offer valuable precedents. The Tree Register of Ireland (TROI), established in 1999 by the Irish Tree Society and the Tree Council of Ireland, aims to document Ireland’s “champion trees”—which are individual specimens distinguished by their exceptional size, age, rarity, or cultural and historical relevance. To date, over 14,000 such trees have been cataloged [85]. In Chile, monumental trees are recognized for their ecological and cultural contributions, including fostering public appreciation for nature, provision of ecosystem services, and value as biocultural heritage. Gutierrez proposed the implementation of a national inventory and the development of policies for their protection [86].

Germany maintains a registry through the German Dendrological Society (DDG), with published works such as those by Kűhn et al. [87]. A similar initiative was launched in 2008 as a joint project between the DDG and the Society German Arboretum (GDA). Since 2009, the register has been continuously updated and made publicly accessible. Since 2010, a “Champion Tree of the Year” has been selected annually and celebrated through local public ceremonies [88].

Databases documenting champion trees also exist in Hong Kong and New Zealand. At the European level, the European Tree of the Year contest—originating from national competitions—raises public awareness and celebrates notable trees. Additionally, the European Champion Tree Forum (ECTF), established in 2010, promotes transnational collaboration. A globally recognized online resource, the Monumental Trees database, provides public access to data on over 30,000 trees worldwide [89].

In Turkey, a standardized methodology for identifying dimensionally significant monumental trees was developed recently. This approach is grounded in data collected through structured oral questionnaires, where participants assigned scores based on the estimated age, height, DBH, crown diameter, and site quality. Observable traits influencing the monumental value were also recorded [90].

##### Technological and Scientific Innovations

In situ measurements (height, crown, diameter) are increasingly combined with geospatial tools. For example, Turkey integrated Geographic Information Systems (GISs) to develop spatial databases and support statistical analyses for better management [91]. In Italy, a “from space to tree” (S2T) approach leverages multi-sensor satellite data to monitor tree health and identify biomechanical risks near monuments [92].

Poland has used acoustic and electrical tomography, along with chlorophyll fluorescence analysis, for comprehensive tree health diagnostics [93]. Similarly, the Resistograph is employed to assess internal decay in monumental specimens [34,94]. Benner [78] used species distribution modeling (Maxent) to map the historical and ecological range of western redcedar (Thuja plicata) using field and archaeological data.

Economic valuation also contributes to conservation. In Sicily, a Contingent Valuation Method (CVM) survey in Madonie Park evaluated the non-material value of monumental hollies and other significant trees [95]. In Apulia, ancient olive groves were assessed using a trunk dimension and shape analysis under Regional Law 14/2007 [96].

##### Guidelines and Conservation Practices

In urban environments, monumental trees—especially those in parks and streets—require consistent health assessments to ensure public safety and tree longevity [115,116]. Large, old trees face increasing threats from climate change, land-use changes, and disturbances [117]. Effective conservation depends on understanding tree longevity and ecological roles [118].

Vegetative propagation enables the preservation of valuable genotypes and ensures genetic continuity [119]. Croatia’s 2022 citizen science campaign to map ancient trees [120] illustrates the power of community involvement. However, comprehensive inventories are lacking in Finland, Norway, and Serbia, unlike in countries such as Germany, France, and the United States [121].

Lindenmayer et al. [9] proposed two primary strategies: (1) reducing mortality/removal of old trees, and (2) facilitating the recruitment and survival of new trees. Conservation methods include in vivo techniques and advanced methods like cryopreservation. In the “Olivo della Strega” case, olive pollen stored in liquid nitrogen retained high viability over a year [122].

Legal frameworks are vital. Spain’s regional acts (Act 4/2004 and Act 14/2006) recognize monumental trees as natural monuments, supporting ecotourism and cultural preservation [123]. Chile and Spain are enhancing legislative protection, including efforts to conserve endangered olive genotypes [124,125].

Long-standing traditions in countries like Austria, Germany, and the UK have led to extensive tree registries. Poland has over 36,000 registered trees; the UK, 22,000; Italy, 4000; France, 2648; Sweden, 1433; Czech Republic, 1300; and Slovakia, 300 [126,127].

Physical interventions are necessary, particularly in cities. Fencing around root zones can minimize trampling and soil compaction [128]. Other maintenance includes crown equilibration, trunk support, and the pruning of diseased or dead branches [129,130,131,132,133,134,135]. Monumental trees are highly vulnerable to pests and diseases (e.g., *Cryphonectria parasitica*, *Ophiostoma novo-ulmi*, *Agrilus planipennis*) [136,137,138,139,140,141,142,143]. Selecting resilient genotypes offers protection against climatic stress and pathogens, especially in urban areas [144,145,146,147,148].

##### Integrated Perspectives and Future Directions

The decline in monumental trees reveals the need for an integrated conservation framework—spanning ecology, policy, and culture [149]. Culturally, features like cavities or gnarled trunks—once seen as defects—are now valued as biodiversity hubs. Conservation in cities is especially complex due to the balance between public safety and heritage preservation [110,150].

Despite advancements, conservation remains uneven. Grassroots campaigns, like those in Croatia [102], demonstrate how community science can bridge institutional gaps. Spain’s legislative models and Chile’s legal proposals reflect increasing global interest in protecting tree heritage.

Inventories must transition from static lists to dynamic, living systems that support policy, research, and education. Europe’s registry networks (e.g., Poland and UK) exemplify the successful integration of ecological and cultural data. However, coverage gaps persist. Bibliometric analyses show the overrepresentation of Europe and North America, leaving Africa, South America, and Asia understudied.

With over 73,000 tree species worldwide, only 45 were identified in this review as monumental—suggesting many more deserve recognition. Conservation frameworks also remain fragmented; legal protections vary greatly between countries, affecting the inclusion of Indigenous and local knowledge.

Cloning resilient trees via grafting—as proposed in Romania [151,152,153,154] and China [155,156,157,158,159,160]—can aid biodiversity and adaptability in reforestation. Yet, most studies focus narrowly on ecological values. Future efforts must integrate myths, traditions, and local histories to preserve the full significance of these living landmarks.

To summarize, future research and action should prioritize the following: expanding geographic representation; developing standardized, interdisciplinary methodologies; integrating cultural and Indigenous knowledge; addressing national and local policy inconsistencies; assessing ecotourism and public awareness impacts; and supporting biodiversity and genetic continuity through propagation and ex situ conservation.

By embracing these dimensions, monumental tree studies can evolve into powerful tools for environmental stewardship, cultural preservation, and sustainable development.

## 3. Materials and Methods

We created a bibliographic database of monumental trees by using the Web of Science (WOS) Core Collection, covering publications from 1 January 1989 to 31 December 2024. This large dataset allowed us to have a holistic overview of research trends and the evolution of studying monumental trees. Scopus was used as a secondary tool. While WOS indexes 7100 journals across 150 scientific disciplines, Scopus covers 16,000 journals, conference proceedings, publications, and books. Both databases have been praised for their coverage, citations, and reliability [161]. For example, Adriaanse and Rensleigh [162] compared the databases and showed that WOS excels at citation counts and journal coverage, while Scopus has more accuracy.

The search terms “monumental trees” or “champion trees” led to 362 articles in WOS and 495 articles in Scopus. We merged these datasets, eliminated the duplicates and entries without abstracts, and retained 204 articles.

Then, our bibliometric study focused on 10 aspects: (1) publication types, (2) Web of Science categories, (3) publication years, (4) geographical distribution, (5) affiliated institutions, (6) publication language, (7) journals, (8) publishers, (9) authorship, and (10) keywords. For the visualization part, we used Microsoft Excel, GeoChart [163,164,165,166], and VOSviewer (version 1.6.20) for the visual mapping and cluster analysis [166].

Our second approach included a traditional review study on the 204 selected articles. Based on this, we grouped the results into three clusters: (1) examples of monumental trees; (2) methods for inventorying monumental trees; and (3) guidelines and methods for the conservation of monumental trees (Figure 9).

We followed the Preferred Reporting Items for Systematic Reviews and Meta-Analyses (PRISMA) guidelines [167]. The selection process of the papers included in this review is illustrated in Figure 10.

## 4. Conclusions

This study offered a comprehensive synthesis of global research on monumental trees by combining bibliometric analysis with a classical review of the scientific literature. The findings underscore the ecological, historical, and cultural significance of monumental trees and the increasing scientific attention they have received, particularly since 2019. Despite this growing interest, several critical gaps persist in terms of the geographic coverage, methodological consistency, and policy implementation.

The bibliometric analysis revealed a concentration of research efforts in Europe and North America, with species such as *Olea europaea* and *Castanea sativa* receiving the most attention. However, the full global diversity of monumental trees remains underdocumented. The variation in terminology and classification criteria across countries highlights the need for standardized definitions to support transnational research and conservation efforts.

Our review of inventory and conservation methods shows promising advances in technology—such as GISs, remote sensing, and non-invasive diagnostics—which are transforming how monumental trees are identified, monitored, and managed. Yet, these tools remain inaccessible or underused in many regions. Moreover, legal protections and conservation programs vary widely, creating uneven outcomes in the preservation of these valuable trees.

Moving forward, conservation strategies must adopt an integrative, interdisciplinary approach that encompasses ecological function, genetic preservation, cultural significance, and public engagement. In addition, international cooperation and standardization will be key to advancing both the scientific understanding and the protection of monumental trees worldwide.

Ultimately, monumental trees are living landmarks—biocultural assets that embody continuity between the past and future. Their preservation requires not only scientific rigor but also social commitment, cultural respect, and long-term policy support. Addressing the identified gaps will be essential to ensure that these irreplaceable natural monuments continue to thrive for generations to come.

## Figures and Tables

**Figure 1 plants-14-02075-f001:**
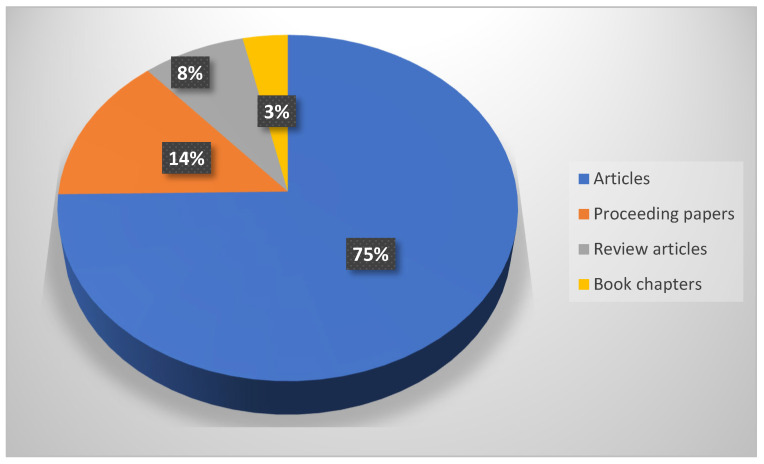
Distribution of publication types among the 204 selected sources on monumental trees, classified according to their designation (article, proceeding paper, review, book chapter) in the Web of Science and Scopus databases.

**Figure 2 plants-14-02075-f002:**
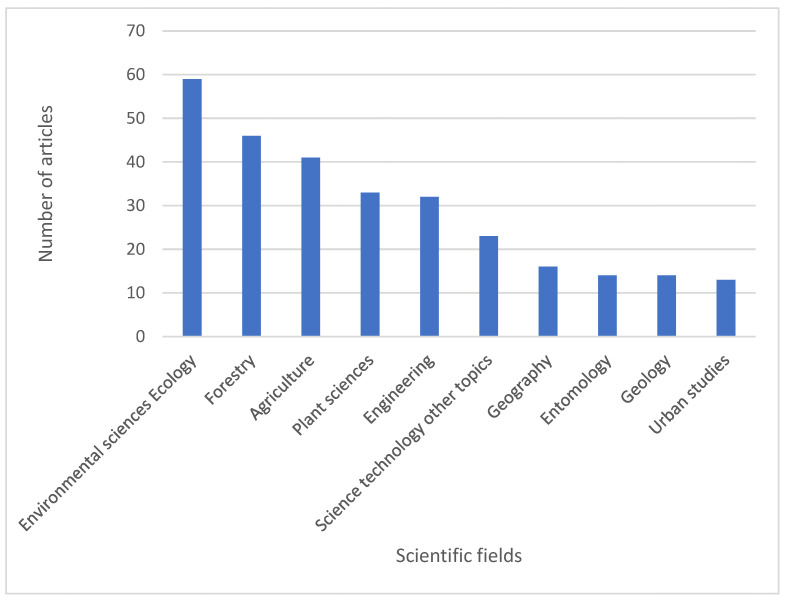
Distribution of the 204 articles on monumental trees across scientific fields, as categorized by Web of Science subject areas.

**Figure 3 plants-14-02075-f003:**
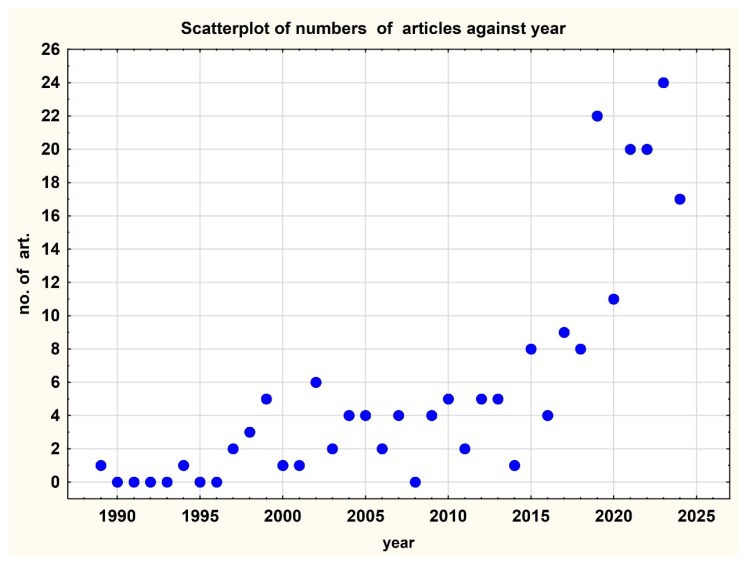
Numbers of articles published per year (1989–2024) in the dataset of 204 publications on monumental trees.

**Figure 4 plants-14-02075-f004:**
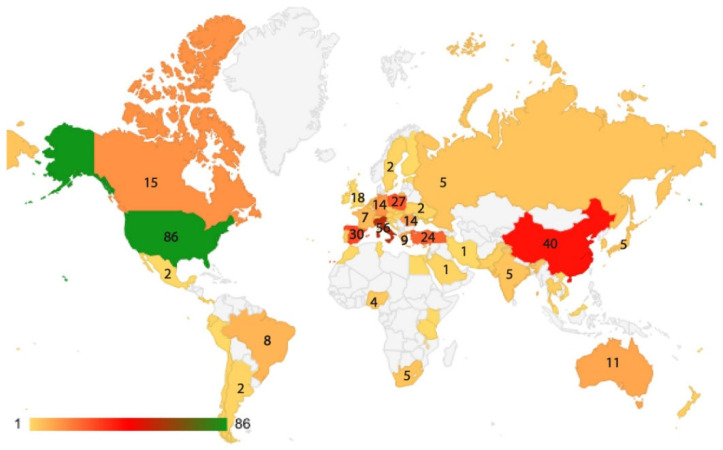
Countries with authors that contributed to articles on monumental trees. The numbers represent the published articles.

**Figure 5 plants-14-02075-f005:**
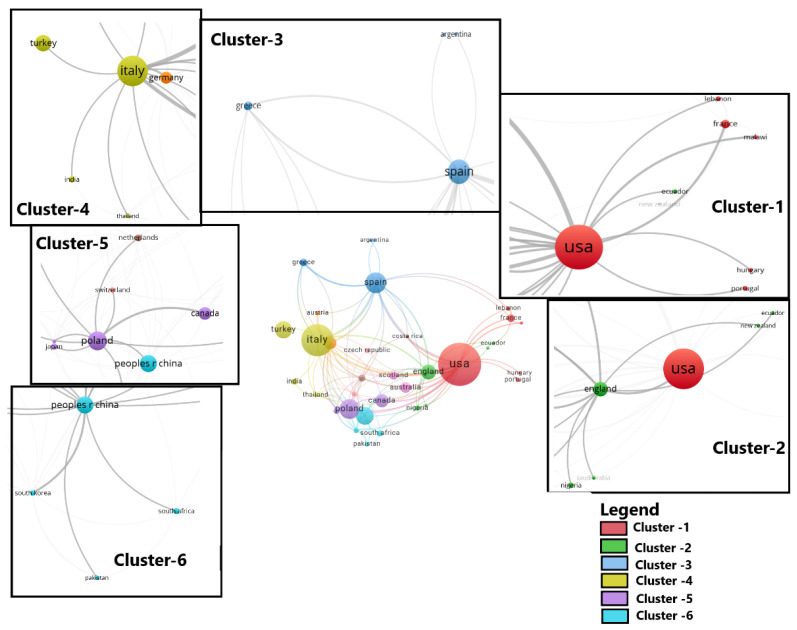
Clusters of countries with authors of articles on monumental trees.

**Figure 6 plants-14-02075-f006:**
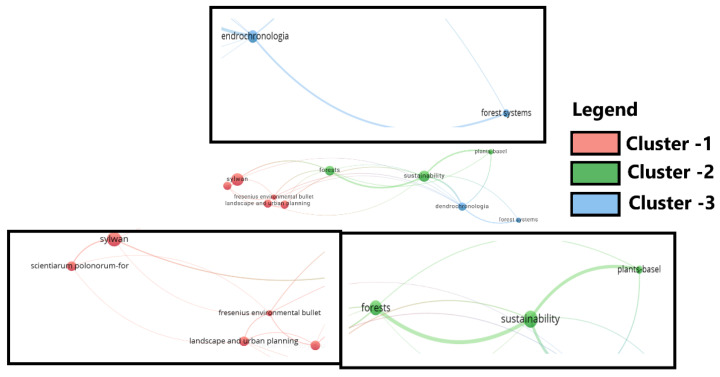
The main journals where articles on monumental trees have been published.

**Figure 7 plants-14-02075-f007:**
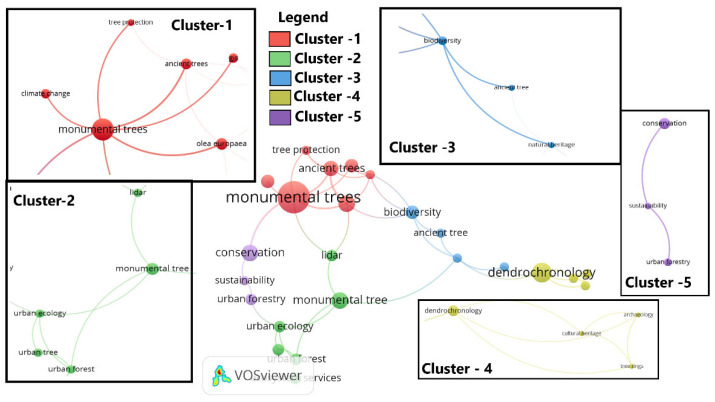
Authors’ keywords concerning monumental trees.

**Figure 8 plants-14-02075-f008:**
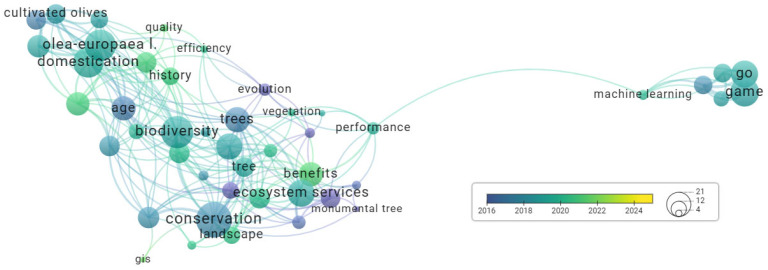
Annual distribution of keywords regarding monumental trees.

**Figure 9 plants-14-02075-f009:**
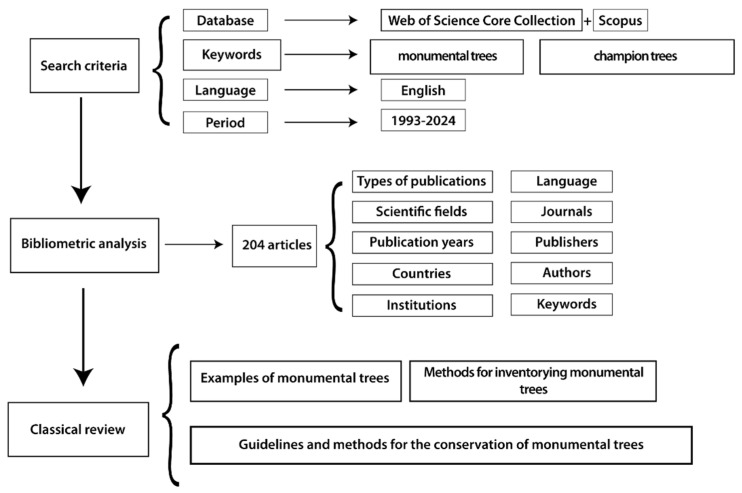
Schematic presentation of the workflow used in our research.

**Figure 10 plants-14-02075-f010:**
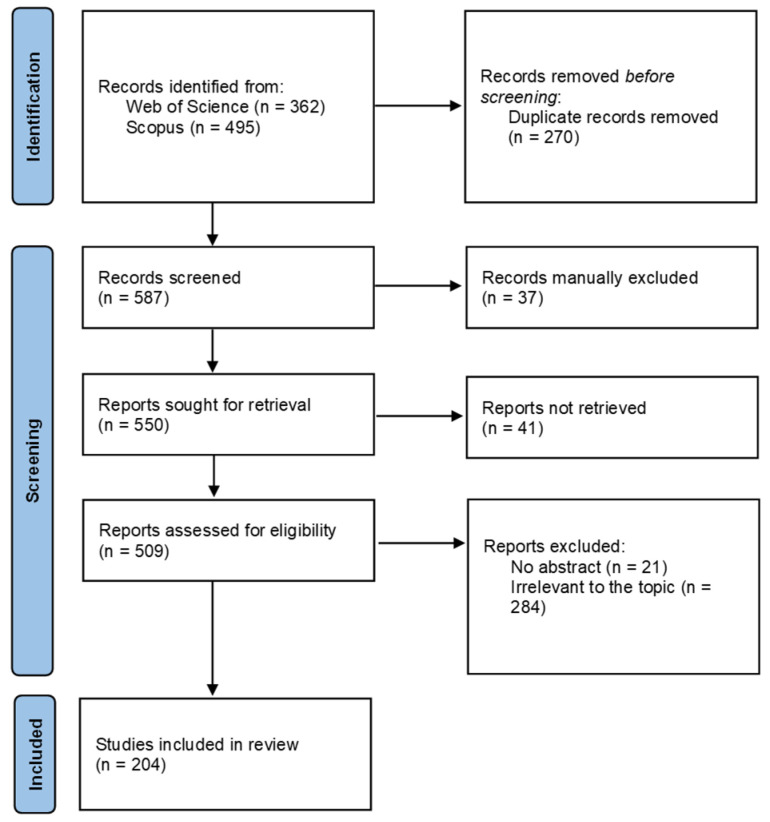
Selection process of the eligible reports based on the PRISMA 2020 flow diagram.

**Table 1 plants-14-02075-t001:** The most frequently used keywords in articles on monumental trees.

Cr. No.	Keyword	Occurrences	Total Link Strength
1	Conservation	20	34
2	Biodiversity	11	30
3	*Olea europaea*	17	29
4	Domestication	8	28
5	Game	14	26
6	Ecosystem services	8	24
7	Management	12	23
8	Trees	11	22
9	Age	8	21
10	Benefits	8	21
11	Genetic diversity	7	19
12	Monumental trees	18	18
13	Ancient trees	6	17
14	Diversity	8	17
15	Dynamics	8	17
16	Cultivated olives	5	16
17	Tree	10	16
18	Urban forest	7	16
19	Landscape	6	14

**Table 2 plants-14-02075-t002:** Monumental trees around the world.

Cur. No.	Monumental Tree Species	Geographical Area	Article Source
1	*Abies alba* Mill.	Garcin, Romania	Vasile et al., 2023 [34]
2	*Abies nordmanniana* (Steven) Spach	Mexico	Mejorado Velazco et al., 2020 [35]
3	*Acer pseudoplatanus* L.	Romania; China	Vasile et al., 2023; Wang et al., 2023 [34,36]
4	*Acer sempervirens* L.	Kuyucak, Turkey	Efe et al., 2014 [37]
5	*Adansonia digitate* L.	Mexico	Mejorado Velazco et al., 2020 [35]
6	*Aesculus glabra* Willd.	Illinois, USA	Ashley et al., 2022 [38]
7	*Aesculus hippocastanum* L.	Brasov, Romania	Vasile et al., 2023 [34]
8	*Ailanthus altissima* (Mill.)	Vienna, Austria	Pisova et al., 2023 [39]
9	*Araucaria angustifolia* (Bertol.) Kuntze	Brazil	Scipioni, 2019 [40]
	*Bombax malabaricum* L.	China	Jim, 2004 [41]
10	*Buxus sempervirens* L.	Vienna, Austria	Pisova et al., 2023 [39]
11	*Carpinus* sp.	Białowieża, Poland	Crzywacz et al., 2018 [42]
12	*Castanea sativa* Mill.	Artvin, Turkey; Brasov, Romania; Switzerland; Spain; Turkey; Italy; Spain	Temel and Ozalp, 2007 [43]; Vasile et al., 2023 [34]; Krebs et al., 2005 [44]; Gaspar Bernárdez Villegas et al., 2021 [45]; Genc and Guner, 2000 [46]; Schicchi et al., 2021 [47]; Fernández-Lorenzo et al., 2024 [48]
13	*Catalpa bignonioides* Walter	Ukraine	Tokarieva et al., 2024 [49]
14	*Ceratonia siliqua* L.	Sicily, Italy	Petino et al., 2024 [50]
15	*Fagus sylvatica* L.	Brasov, Romania; Sonian Forest, Belgium	Vasile et al., 2023 [34]; Verschuren et al., 2023 [51]
16	*Ficus microcarpa* L. f.	Shenzhen, China	Lai et al., 2019 [52]
17	*Fraxinus* sp.	Białowieża, Poland	Crzywacz et al., 2018 [42]
18	*Fraxinus udhei* (Wenz.) Lingelsh	Mexico	Mejorado Velazco et al., 2020 [35]
19	*Ginkgo biloba* L.	Shenzhen, China; Brasov, Romania	Lai et al., 2019 [52]; Li and Zhang, 2021 [53]; Vasile et al., 2023 [25]
20	*Gleditsia triacanthos* L.	Vienna, Austria	Pisova et al., 2023 [39]
	*Juniperus excelsa* Bieb.	Turkey	Kurtaslan et al., 2012 [54]
	*Juniperus przewalskii* Kom.	China	Liu et al., 2019 [55]
21	*Juniperus thurifera* L.	Soria, Spain	Fuster and Sadornil, 2020 [56]
22	*Larix decidua* Mill.	Poland	Meller and Bernat, 2019 [57]
23	*Metasequoia glyptostroboides* Hu and W.C.Cheng, 1948	Ukraine	Tokarieva et al., 2024 [49]
24	*Ocotea porosa* (Nees & Martius) Barroso	Brazil	Scipioni, 2019 [40]
25	*Olea europaea* L.	Bchaaleh, Lebanon; Spain; Sicily, Italy; Crete, Greece; Cyprus	Camarero et al., 2024 [58]; Arnan et al., 2012 [59]; Marchese et al., 2023 [60]; Bombarely et al., 2021 [61]; Anestiadou et al., 2017 [62]
	*Phoebe zhennan* S. Lee & F.N. Wei	China	Yang et al., 2024 [63]
26	*Picea abies* L. Karst	Białowieża Primeval Forest, Poland	Nowakowska et al., 2020 [64]
27	*Pinus nigra* subsp. pallasiana	Elemen Plateau of Dirgine in Zonguldak, Turkey	Yaman and Sarıbaș, 2007 [65]
28	*Pinus nigra* J.F. Arnold subsp. calabrica (Poir.)	Fallistro, Italy	Bernardini et al., 2020 [66]
29	*Platanus hybrida* L.	Pisa, Italy; Spain	Marchica et al., 2017 [67]; Millan et al., 2025 [68]
30	*Platanus orientalis* L.	Istanbul, Turkey; Greece	Yener, 2022 [69]; Kagiali and Tsitsoni, 2019 [70]; Grigoriadis et al., 2021 [71]
31	Populus nigra L.	Jaworze, Poland	Pietraszko et al., 2022 [72]
32	*Pyrus communis L.*	Poland	Kimic, 2021 [73]
33	*Quercus infectoria*	Edremit, Turkey	Efe et al., 2011 [74]
34	Quercus robur L.	Jasienica, Poland	Pietraszko et al., 2022 [72]
35	*Quercus hartwissiana* Stev.	Turkey	Gul et al., 1999 [75]
36	*Rhododendron myrtifolium* Schott & Kotschy, 1851	Ukraine	Tokarieva et al., 2024 [49]
37	*Sambucus nigra L.*	Vienna, Austria	Pisova et al., 2023 [39]
38	*Sequoiadendron giganteum* (Lindl.) J.Buchh, 1939	Czech Republic	Dreslerova, 2017 [15]
39	*Sorbus torminalis* (L.) Crantz	Kyiv, Ukraine	Prokopuk et al., 2022 [76]
40	*Taxodium Mucronatum* Ten.	Mexico	Mejorado Velazco et al., 2020 [35]
41	Taxus baccata L.	Thasos, Greece; Jaworze, Poland;	Malliarou et al., 2023 [77]; Pietraszko et al., 2022 [72]
42	*Taxus chinensis* (Rehder & E.H.Wilson) Rehder	Shenzhen, China	Lai et al., 2019 [52]
43	*Thuja plicata* Donn ex D. Don	Canada	Benner et al., 2019 [78]
44	Tilia cordata Mill.	Jasienica, Poland	Pietraszko et al., 2022 [72]
45	*Tilia platyphyllos* Scop.	Czech Republic	Dreslerova, 2017 [15]
	*Ziziphus jujuba* Mill.	China	Pan et al., 2025 [79]

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
