# Peer review of "Living Landmarks: A Review of Monumental Trees and Their Role in Ecosystems"

_plants, 2025, doi:10.3390/plants14132075_

Round 1
Reviewer 1 Report
Comments and Suggestions for Authors
The MS only synthesizes the available global research in English about monumental trees. In fact, the most literatures on the monumental trees were published in native language. I have search the monumental tree in Chinese CNKI Literature Library and got 3,849 articles, 6 Ph.D. dissertations and 163 Master's thesis. All of them have the English abstract. The MS need a major revision for summarizing more literatures.
Author Response
Comments 1
The MS only synthesizes the available global research in English about monumental trees. In fact, the most literatures on the monumental trees were published in native language. I have search the monumental tree in Chinese CNKI Literature Library and got 3,849 articles, 6 Ph.D. dissertations and 163 Master's thesis. All of them have the English abstract. The MS need a major revision for summarizing more literatures.
Response 1
Thank you for your valuable comment and for highlighting the importance of including literature published in native languages. Following a more in-depth review of the literature, including sources in native languages, we have added 58 additional references to the manuscript and updated the text accordingly. We believe that, with these additions, the content of our article is now more comprehensive, balanced, and representative of the global research on monumental trees.
Reviewer 2 Report
Comments and Suggestions for Authors
The manuscript examines the important topic of monumental trees from a biological and social perspective. The review article has many advantages, but there are also some shortcomings. Without going into detail about the advantages, I would like to point out the most important shortcomings and parts of the manuscript that need to be corrected.
- I recommend improving the abstract and making it more specific. The current text is more promotional in nature. For example, the problem should be raised at the beginning, and a specific message to the reader should be included at the end of the abstract. Overly general statements that do not allow conclusions to be drawn about the content of the article should be avoided in the abstract.
- I believe that the text of the manuscript should be written in a more precise and academic style, rather than in an advertising or slogan style. From the beginning to the end of the manuscript, there are numerous environmental clichés that are often far from the truth or inaccurate. For example, "Their contribution is essential in influencing and regulating environmental factors." (lines 42-43). There is a lack of precision: what specific environmental factors do they influence? The manuscript is full of such overly generalised sentences that say nothing specific. Precision is important in academic writing.
- There is a lack of precision in the results. For example, the caption for Figure 1 does not indicate the criteria used for the distribution. What are the criteria for the categorisation? The caption for Figure 2 should be edited entirely. It does not present the distribution of scientific fields, but rather an analysis of the distribution of articles by scientific field. Similar shortcomings are found throughout the text. The caption for Figure 3 is even more inaccurate. The number of articles should be specified. In addition, the presentation type is inappropriate. Such data cannot be presented in a curve; columns or unconnected points are more suitable.
- The color scale in Figure 4 is not good because the map is unclear. It is almost impossible to distinguish between the differences in the prints, let alone identify the meaning of the print. For example, how much does the number of authors in France differ from that in Spain? How was the number of authors determined? Since the methodology is at the end of the article, it is not possible to clearly understand the meaning of the information from the text and illustrations.
- In Figures 5-8, the color scale is also inadequate because the differences between shades are very subtle. Furthermore, what is the theoretical and practical significance of keyword analysis by year (Figure 8)? I did not find this in the article. What do we learn from such an analysis? I am convinced that the results should provide information with a clear purpose. Information that is beautiful but has no purpose is just noise.
- There cannot be sections consisting solely of tables (2.1.1). A short sentence referring to a table without any analysis is not sufficient.
- The structure of the text in subchapter 2.1.2. is unclear. Why are the lower-level sections of the text not numbered? (lines 169, 199, etc.).
- In my opinion, the biggest drawback of this review article is that it includes a discussion section, the content of which is essentially the same as the analysis of sources. It is very difficult to see how the results differ from the discussion. Therefore, I recommend that the authors consider whether the structure of a research article is really necessary for a review article. It would be better to divide the content according to the topics discussed and discuss everything in detail in one place, rather than dividing the information into several parts.
The language is not bad, but it needs to be more precise and clear. Some of the text contains technical jargon that not all readers will interpret in the same way.
Author Response
Comments 1
I recommend improving the abstract and making it more specific. The current text is more promotional in nature. For example, the problem should be raised at the beginning, and a specific message to the reader should be included at the end of the abstract. Overly general statements that do not allow conclusions to be drawn about the content of the article should be avoided in the abstract.
Response 1
Thank you for your insightful suggestion regarding the abstract. We have revised it to clearly introduce the problem at the beginning, focus on specific findings from our analysis, and provide a clear concluding message. The revised abstract avoids general statements and highlights the challenges faced by monumental trees, the key research trends, methodologies, and the necessity of multidisciplinary conservation approaches. We believe this version better reflects the content and significance of our study and addresses your concerns.
Comments 2
I believe that the text of the manuscript should be written in a more precise and academic style, rather than in an advertising or slogan style. From the beginning to the end of the manuscript, there are numerous environmental clichés that are often far from the truth or inaccurate. For example, "Their contribution is essential in influencing and regulating environmental factors." (lines 42-43). There is a lack of precision: what specific environmental factors do they influence? The manuscript is full of such overly generalised sentences that say nothing specific. Precision is important in academic writing.
Response 2
We thank the reviewer for this valuable observation. In response, we have carefully revised the entire article to eliminate vague or overly general expressions and improve academic precision. For instance, we replaced the sentence “Their contribution is essential in influencing and regulating environmental factors” with a more specific formulation: “Their ecological role includes influencing and regulating various environmental factors such as microclimate, air quality, soil stability, and biodiversity.” Similar changes were made throughout the article to clarify meanings, provide concrete examples, and ensure a scientific tone consistent with academic standards. We believe these revisions have significantly strengthened the clarity, rigor, and credibility of the manuscript.
Comments 3
There is a lack of precision in the results. For example, the caption for Figure 1 does not indicate the criteria used for the distribution. What are the criteria for the categorisation? The caption for Figure 2 should be edited entirely. It does not present the distribution of scientific fields, but rather an analysis of the distribution of articles by scientific field. Similar shortcomings are found throughout the text. The caption for Figure 3 is even more inaccurate. The number of articles should be specified. In addition, the presentation type is inappropriate. Such data cannot be presented in a curve; columns or unconnected points are more suitable.
Response 3
We thank the reviewer for this constructive and helpful observation. In response:
- Figure 1: We have revised the caption to clarify that the publication types (article, proceeding paper, review, book chapter) were classified according to their indexing in the Web of Science and Scopus databases. The updated caption now reads:
"Figure 1. Distribution of publication types among the 204 selected sources on monumental trees, classified according to their designation in the Web of Science and Scopus databases." - Figure 2: We have updated the caption to more accurately reflect the content. It now states:
"Figure 2. Distribution of the 204 articles on monumental trees across scientific fields, as categorized by Web of Science subject areas." - Figure 3: We agree with the reviewer that the previous presentation format was not appropriate for this type of data. We have:
- Replaced the curve graph with a scatterplot chart showing the number of articles per year.
- Revised the caption to:
"Figure 3. Number of articles published per year (1989–2024) in the dataset of 204 publications on monumental trees."
We have also reviewed the rest of the manuscript to ensure that figure captions and result descriptions are accurate and precise. Thank you again for helping us improve the clarity of our work.
Comments 4
The color scale in Figure 4 is not good because the map is unclear. It is almost impossible to distinguish between the differences in the prints, let alone identify the meaning of the print. For example, how much does the number of authors in France differ from that in Spain? How was the number of authors determined? Since the methodology is at the end of the article, it is not possible to clearly understand the meaning of the information from the text and illustrations.
Response 4
Thank you for your valuable feedback. We have revised Figure 4 by updating the color scale to ensure greater visual contrast and clarity. The colors now clearly reflect the number of publications, making it easier to distinguish between countries. Additionally, we have included the actual number of articles next to the majority of the countries for better interpretability.
Comments 5
In Figures 5-8, the color scale is also inadequate because the differences between shades are very subtle. Furthermore, what is the theoretical and practical significance of keyword analysis by year (Figure 8)? I did not find this in the article. What do we learn from such an analysis? I am convinced that the results should provide information with a clear purpose. Information that is beautiful but has no purpose is just noise.
Response 5
We appreciate the reviewer’s valuable feedback regarding the graphical clarity and the need for a more explicit interpretation of the keyword trend analysis.
- Color Scale in Figures 5–8:
We acknowledge that the previous color palette lacked sufficient contrast. All four figures (Figures 5–8) have been updated using a revised color scheme with enhanced contrast between clusters and keyword categories, ensuring better visual distinction and readability. - Theoretical and Practical Significance of Figure 8:
We thank the reviewer for this important observation. In response, we have: - Added a new explanatory paragraph in the Results section (Section 2.1) to clarify the relevance of temporal keyword trends.
- This analysis helps to identify shifts in research focus over time, revealing how interest in certain topics—such as “urban forest,” “biodiversity,” or “Olea europaea”—has evolved. These patterns reflect changing scientific priorities, funding directions, and societal concerns related to monumental trees.
- For example, the increase in terms like “conservation” and “management” after 2019 aligns with global initiatives on biodiversity preservation and climate resilience.
- Practically, such analysis informs future research directions, helps in identifying gaps, and guides interdisciplinary collaboration.
We have made the purpose and interpretation of Figure 8 explicit in the revised manuscript to avoid any ambiguity and ensure that it contributes meaningfully to the overall narrative.
Comments 6
There cannot be sections consisting solely of tables (2.1.1). A short sentence referring to a table without any analysis is not sufficient.
Response 6
Thank you for your observation. We have revised Section 2.1.1 to include a narrative analysis that contextualizes the data presented in Table 2. The revised section now highlights key patterns, such as geographical distribution, frequently cited species, and the ecological and cultural significance of monumental trees. This addition aims to provide a clearer interpretation of the diversity and importance of the species listed, in accordance with your suggestion.
Comments 7
The structure of the text in subchapter 2.1.2. is unclear. Why are the lower-level sections of the text not numbered? (lines 169, 199, etc.).
Response 7
Thank you for pointing this out. We corrected the numbering error where subchapter 2.1.3 was mistakenly labeled as 2.2.3. Additionally, we have now clearly numbered the subsections under 2.1.2 for consistency and clarity. The updated structure is as follows:
- 2.1.2.1. National and regional inventory approaches
- 2.1.2.2. Advances in technological and scientific methods
We believe this improves the readability and organization of the text.
Comments 8
In my opinion, the biggest drawback of this review article is that it includes a discussion section, the content of which is essentially the same as the analysis of sources. It is very difficult to see how the results differ from the discussion. Therefore, I recommend that the authors consider whether the structure of a research article is really necessary for a review article. It would be better to divide the content according to the topics discussed and discuss everything in detail in one place, rather than dividing the information into several parts.
Response 8
We thank the reviewer for the insightful comment regarding the similarity in content between the Results and Discussion sections and the suggestion to consider a more integrated structure, typical of narrative reviews.
While we fully understand and appreciate this perspective, we are not able to change the article’s structure, as the journal requires separate Results and Discussion sections to maintain a standardized scientific format. However, in response to the reviewer’s valid concern, we have carefully revised both pairs of sections—2.1.2 / 3.3 and 2.1.3 / 3.4—to ensure a clearer and more rigorous distinction between descriptive content (Results) and interpretative analysis (Discussion).
- In Section 2.1.2 (Methods for inventorying monumental trees), we now focus exclusively on factual reporting of national and regional approaches, technical criteria, databases, and tools used across different countries and regions.
- In Section 3.3 (Discussion on inventory methods), we interpret these findings in the context of cultural, ecological, and socio-political implications, highlighting trends, challenges, and the importance of evolving from size-based systems to more integrative frameworks.
- Similarly, in Section 2.1.3 (Guidelines and methods for the conservation of monumental trees), we present a concise and structured synthesis of conservation strategies, legal instruments, biological threats, and technological methods as documented in the literature.
- In Section 3.4 (Discussion on conservation methods), we build on this evidence to explore its broader meaning, analyze gaps and inconsistencies in global practices, and propose future directions, including interdisciplinary and culturally informed approaches.
We hope these changes clarify the manuscript’s structure and improve its coherence while remaining within the format required by the journal. We are grateful for the reviewer’s helpful feedback, which contributed to strengthening the manuscript.
Round 2
Reviewer 1 Report
Comments and Suggestions for Authors
Fig. 5 The bottom lines of Cluster7 & 8 are lost.
Fig. 7 What is it mean for " VOSviewer" rectangle?
Author Response
Comments 1
Fig. 5 The bottom lines of Cluster7 & 8 are lost.
Response 1
Thank you for your observation. We have removed Clusters 7 and 8 from Figure 5, as they were not central to the discussion and caused unnecessary visual congestion. The updated figure is now clearer and more focused, enhancing readability and interpretability.
Comments 2
Fig. 7 What is it mean for " VOSviewer" rectangle?
Response 2
We appreciate your question. The rectangle with the label "VOSviewer" is automatically generated by the VOSviewer software as part of the visual output. It serves as a watermark or attribution to the tool used for network visualization. Unfortunately, the software does not currently allow the removal of this label. We have clarified this in the figure caption to avoid confusion for readers.
Reviewer 2 Report
Comments and Suggestions for Authors
The authors took most of the comments into account or provided reasoned responses. The quality of the revised manuscript has been significantly improved, although a few minor shortcomings remain.
- Regarding the color scale. I understand very well that some software does not offer other options. However, it is important to understand that there are many scientists with visual impairments. If a person without impairments has difficulty distinguishing shades in an illustration on a screen, how will it be possible to distinguish them when the text is printed? This is a rhetorical question that authors should consider in the future and look for graphic expression tools that would allow all readers to understand the meaning of illustrations.
- I disagree with the authors' position on comment 8 in the first review. I would like to point out that they did not read the recommendations for the structure of review articles carefully. It is clearly defined in the "Instructions for authors" (https://www.mdpi.com/journal/plants/instructions). If the authors had paid attention to the instructions, it would not have been necessary to reorganize the entire manuscript several times.
Author Response
Comments 1
Regarding the color scale. I understand very well that some software does not offer other options. However, it is important to understand that there are many scientists with visual impairments. If a person without impairments has difficulty distinguishing shades in an illustration on a screen, how will it be possible to distinguish them when the text is printed? This is a rhetorical question that authors should consider in the future and look for graphic expression tools that would allow all readers to understand the meaning of illustrations.
Response 1
Thank you for highlighting this important issue. We fully acknowledge the need for inclusive and accessible visual design. While the current version of the software limits our customization options, we will prioritize the use of more accessible color schemes in future studies and seek tools that support better visual accessibility for all readers, including those with visual impairments.
Comments 2
I disagree with the authors' position on comment 8 in the first review. I would like to point out that they did not read the recommendations for the structure of review articles carefully. It is clearly defined in the "Instructions for authors" (https://www.mdpi.com/journal/plants/instructions). If the authors had paid attention to the instructions, it would not have been necessary to reorganize the entire manuscript several times.
Response 2
Thank you for your constructive criticism. We sincerely apologize for the oversight and have now carefully reviewed the "Instructions for Authors" for Plants. Based on your guidance, we have restructured the manuscript accordingly, merging the Results and Discussion sections where appropriate. Your comment was instrumental in improving the clarity and alignment of the manuscript with the journal's standards, and we are grateful for your attention to detail.
Round 3
Reviewer 2 Report
Comments and Suggestions for Authors
I have no further comments and recommend that the manuscript be accepted.